# LNC-ing Genetics in Mitochondrial Disease

**DOI:** 10.3390/ncrna10060057

**Published:** 2024-11-15

**Authors:** Rick Kamps, Emma Louise Robinson

**Affiliations:** 1Department of Translational Genomics, School for Mental Health and Neuroscience (MHeNS), Maastricht University, P.O. Box 616, 6200 MD Maastricht, The Netherlands; 2Department of Medicine, Division of Cardiology, University of Colorado Anschutz Medical Campus, Aurora, CO 80045, USA; emma.2.robinson@cuanschutz.edu

**Keywords:** mitochondrial disease (MD), long non-coding RNA (lncRNA), genome sequencing (GS), disease coding and non-coding variants, multiomics, biomarkers

## Abstract

Primary mitochondrial disease (MD) is a group of rare genetic diseases reported to have a prevalence of 1:5000 and is currently without a cure. This group of diseases includes mitochondrial encephalopathy, lactic acidosis, and stroke-like episodes (MELAS), maternally inherited diabetes and deafness (MIDD), Leber’s hereditary optic neuropathy (LHON), Leigh syndrome (LS), Kearns–Sayre syndrome (KSS), and myoclonic epilepsy and ragged-red fiber disease (MERRF). Additionally, secondary mitochondrial dysfunction has been implicated in the most common current causes of mortality and morbidity, including cardiovascular disease (CVD) and cancer. Identifying key genetic contributors to both MD and secondary mitochondrial dysfunction may guide clinicians to assess the most effective treatment course and prognosis, as well as informing family members of any hereditary risk of disease transmission. Identifying underlying genetic causes of primary and secondary MD involves either genome sequencing (GS) or small targeted panel analysis of known disease-causing nuclear- or mitochondrial genes coding for mitochondria-related proteins. Due to advances in GS, the importance of long non-coding RNA (lncRNA) as functional contributors to the pathophysiology of MD is being unveiled. A limited number of studies have thus far reported the importance of lncRNAs in relation to MD causation and progression, and we are entering a new area of attention for clinical geneticists in specific rare malignancies. This commentary provides an overview of what is known about the role of lncRNAs as genetic and molecular contributors to disease pathophysiology and highlights an unmet need for a deeper understanding of mitochondrial dysfunction in serious human disease burdens.

## 1. Introduction

The revolution in genomic sequencing technologies of the last decennia has revealed that the ~1.5% of the protein-coding portion of the human genome (~21,000 genes) has important implications in human diseases. Primary mitochondrial disease (MD) is a group of rare diseases which pose heterogeneous phenotypical threats to public health worldwide. They have a genetic basis caused by mutations either in the mitochondrial DNA (mtDNA) or nuclear DNA (nDNA), directly impacting the mitochondria’s ability to produce energy. Primary MD is reported to have a prevalence of 1:5000 and is currently without a cure. This group of diseases includes mitochondrial encephalopathy, lactic acidosis, and stroke-like episodes (MELAS), maternally inherited diabetes and deafness (MIDD), Leber’s hereditary optic neuropathy (LHON), Leigh syndrome (LS), Kearns–Sayre syndrome (KSS), and myoclonic epilepsy and ragged-red fiber disease (MERRF). In contrast to the relatively low prevalence of primary MD, secondary mitochondrial dysfunction has been implicated in the most common current causes of mortality and morbidity, including cardiovascular disease (CVD) and cancer. Identifying key genetic contributors to both MD and secondary mitochondrial dysfunction may guide clinicians to identify the most effective treatment course and prognosis, reveal new therapeutic drug targets, as well as inform family members of any hereditary risk of disease transmission.

Clinical reports show that MD can display multiple symptoms at any time and at any age. Each human cell contains hundreds to thousands of individual mitochondria, depending on the ATP demands of that cell type. It is possible for a single mutation to lead to disease if the mutation levels are above the threshold (>70%) in specific energy-rich tissues and organs. The combination of mutated and wild-type mtDNA is called heteroplasmic (1–95%) or homoplasmic levels (96–100%), which contribute to this heterogeneous progressive nature of primary MD. Profiling these mutation levels can be complicated, since the relative abundance of the mutated mtDNA is variable in specific primary cases such as MELAS, MERRF, LHON, MIDD, LS, and KSS. The complexity of MD is not only the primary extensive evaluation of the biochemistry, immunostainings, and examination of clinical symptoms but also the complexity in stability and mitochondrial maintenance (homeostasis). There are two independent regulatory and genetic mechanisms related to the transmission and occurrence of MD. The inherited nuclear defects in nuclear genomic DNA (nDNA) may be inherited as autosomal recessive or dominant mutations or X-linked, but also as “de novo” mutations. During early oogenesis and after fertilization, the embryogenesis processes a mitochondrial bottleneck, showing a minimum number of copies of mitochondria per oocyte. But also, positive and negative mutation selection, or genetic drift, is poorly understood in this stage of embryos. All these different mechanisms influence the method of choice in unique MD treatment.

Advances in deep RNA-sequencing technologies have revealed more than 90% of the human genome to be transcribed to RNA, despite only around 1.5% coding for proteins [1]. Progress in characterizing cell type and temporal-specific expression patterns and molecular functionality of some of these non-coding RNA species has resulted in the renaming of this previously named “dark matter” of the genome to the “non-coding genome”. One significant family of non-coding RNAs are the long non-coding RNAs (lncRNAs), which are defined as non-coding RNAs of more than 200 nucleotides (nt) in length. Like protein-coding genes, lncRNAs can be expressed as different isoforms. Some have introns and exons and undergo post-transcriptional processing, and splice variants exist. LncRNA genes are present intragenically within intronic regions of protein-coding genes, overlapping in the sense or antisense direction to protein-coding genes or other lncRNAs. In the case of overlapping genes, the expression and regulation of the lncRNA can be independent of that of the associated protein-coding gene [2]. LncRNAs are distinct from the short (21–23 nt) post-transcriptional gene repressor microRNAs, which repress RNA transcription through complementary base pairing to their targets. Whilst human mtDNA encodes at least 13 microRNAs, this commentary will focus on lncRNAs, which have a broader spectrum of potential functional modalities in mitochondrial biology, and there is a greater knowledge gap in the context of MD and mitochondrial dysfunction [3]. The human genome reveals approximately ~270,044 unique lncRNA transcripts, although up to 80% of these remain uncharacterized in relation to disease causation and progression [4]. LncRNA genes (~20,000) have a dual genetic origin, with a subset transcribed from the mitochondrial genome (mt-lncRNAs), with the remainder encoded by the nDNA genome, although a different biological mechanism and function [5]. The latest sequencing technologies as long-read (LR) platforms are more sensitive detection methods using a level of 10× coverage to determine the complete characterization of the lncRNA as the non-coding genome. LR also improves the levels of these poor conservative regions or, in case of low copy numbers, biological gene structures and functional features that are partly missed in short-read sequencing (SR) due to mapping or sequencing issues. A significant number of disease-associated lncRNAs have been characterized thus far, based upon their patterns of expression linked to a specific region in the human genome [5]. The variety of lncRNAs, such as that of known proteins, seems not to be limited to their biological functions as protein regulators, which are remodeling chromatin or guide molecules in the nucleus. But also, as antisense inhibitors that block transcription, as primary and secondary regulators, or as co-factors of proteins involved in two-way directional nuclear transport, this mechanism remains largely unknown in the literature. Recently, especially lncRNAs are relevant as RNA transcripts, which highlighted specific regular and cellular communication in standard cell physiology [6].

## 2. Genomics Approaches to Identifying Mediators of Disease

In the last decennia, next-generation sequencing (NGS) has evolved as a state-of-the-art technology to unravel the pathophysiology of disease-causing exonic variants in MD. MD are the most common genetic metabolic diseases, currently affecting approximately 1 in 5000 individuals, excluding the variants in the “non-coding genome” or long non-coding RNAs (lncRNAs). The limitation of exome sequencing (ES) in combination with SR is already outdated as the preferred first successful method, and it still leaves 31% of patients undiagnosed [7]. The introduced trio analysis of GS is now the more preferred method based on allelic origin for nDNA variants (maternal, paternal) and mitochondrial DNA variants (maternal mtDNA, Figure 1) occurs, or if “de novo”, which rapidly reduces the total number of variants that require interpretation in bioinformatics filtered ES/GS dataset. Additional functional analysis is conducted in undiagnosed cases or in variants of unknown genes applying or to guide the genomic quest in specialized techniques, e.g., whole transcriptomics, proteomics, or combined metabolomics, known collectively as multiomics. In summary, these techniques are contributing to the level of diagnostic yield in unraveling the “missing-lnc” as a genetic cause of MD development [6,8].

Improved depth and sensitivity of diagnostic techniques (e.g., long-read whole genome sequencing, optical genome mapping for determining chromosomal and structural variants, and epigenetic profiling for screening methylation sites) will further enhance the diagnostic yield for MD [8]. Using long-read (LR) genome sequencing techniques with lncRNAs show fragments as >200 nt with an average length of ~1000 nt will be the next era in genomics that could unravel the genetic component of undiagnosed MD, which is formulated as idiopathic or as “de novo” in case of related disease-causing variants affecting the proteins [6]. In general, more research needs to be conducted to understand the exact function of ncDNA in specific conserved regions in the multiple species (animal models), which are also listed as the untranslated regions as UTRs, as canonical splice and promoter sites, or as short-long ncRNA disease mechanism involved in humans [8,9,10].

**Figure 1 ncrna-10-00057-f001:**
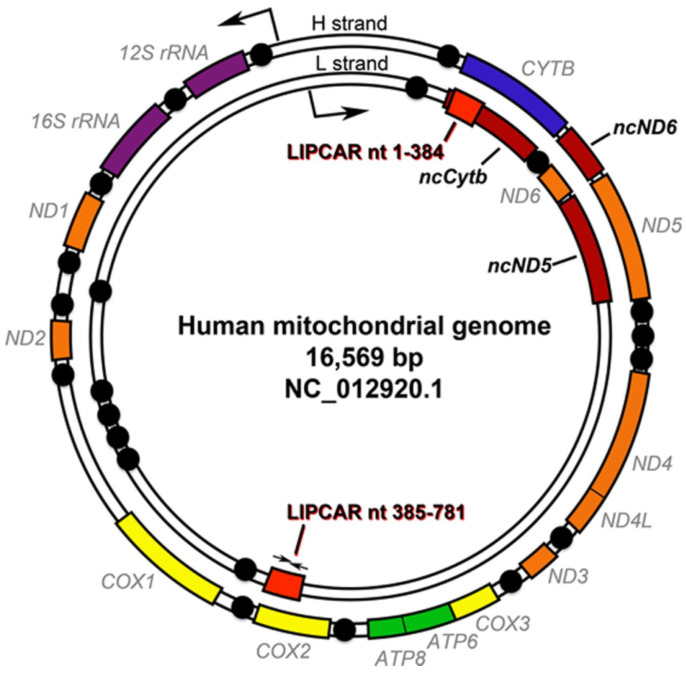
The mitochondrial DNA (mtDNA) and location of ncRNAs. The mtDNA consists of 16,569 base pairs, which encode 22 tRNAs, 2 rRNAs, and the 13 core subunits of the OXPHOS complexes. Here, the complex I genes of interest are in yellow, the complex III genes are in blue, the complex IV genes are shown in orange, and the complex V genes are in green. The positions of the two rRNAs are in purple, and the 22 single tRNAs are colored in black. Known validated mt-lncRNAs are listed as dark orange regions. Here, as an example, the LIPCAR sequence maps to the two red segments as small arrows indicating the selected designed standard primers. Heavy indicates the H-chain, and the Light one is the L-chain. The image is reprinted/adapted with permission from Gerald W. Dorn. Circulation Research. LIPCAR, Volume: 114, Issue: 10, Pages: 1548–1550, DOI: (10.1161/CIRCRESAHA.114.304028), © 2014 American Heart Association, Inc. [11].

## 3. Secondary Mitochondrial Dysfunction and LncRNAs

The role of lncRNAs in secondary mitochondrial dysfunction and cardiovascular diseases is emerging. In a 2014 study, deep RNA-seq from end-stage cardiomyopathy biopsies revealed strong enrichment of mitochondrial RNA, with 71% of the lncRNA reads originating from nine mitochondrial lncRNAs. Gene expression was analyzed from biopsies in failing hearts before and after left ventricular assisted device (LVAD) implantations. Altered heart function has been linked to changes in the number of copies of mitochondria (mtDNA copy number) and dynamics in mitochondrial morphology under pathophysiological conditions. These mitochondria-encoded lncRNAs are related to stress response [12]. Future studies are needed to delineate the function of these lncRNAs in the heart and their mechanisms of action to provide more potential insights into lncRNA biology in relation to cardiovascular diseases [12]. LncRNAs are established as important biological mediators in disease, but the translation of this knowledge to implement changes in the clinical genetics diagnostics lab is still lacking.

The molecular and chemical properties of RNA species give them a unique druggability that proteins do not have. Their nucleotide sequences render them open to sensitive and specific inhibition using antisense technology by designing nucleic acid-based therapeutics complementary in base sequence to the intended RNA target [6]. Few studies targeting ncRNAs have extended beyond the preclinical stage. One exception is the microRNA antisense oligonucleotide drug CDR132L, which is currently in Phase II clinical trials for ischemic heart failure [13]. However, research in mitochondrial mt-lncRNAs is accelerating in the field of ncRNAs. During transcription of the two mito-strands, the heavy and the light chains of mitochondrial DNA, mainly the light chain carries only seven tRNA genes and the “solo” protein-encoded *ND6* gene, which generates the ND6 protein and releases large non-coding sequences (e.g., lncRNA) during transcriptional processing shown in Figure 1. Essentially, the level of mt-lncRNAs regulating processes as inflammation, or mediated as metastasis, and the level of apoptosis induction. The sense non-coding mitochondrial RNAs, SncmtRNA, antisense non-coding mitochondrial RNA, and ASncmtRNA play an important role in ubiquitous cell cycle regulation within cancer cells. In different types of cancer cells, these ASncmtRNAs are downregulated and normally expressed in proliferating cells [14]. Also, the identification of lncND5, lncCyt b, and lncND6 as lncRNAs in purified mitochondria analyses in strand-specific RNA-seq data revealing RNA-RNA duplexes involved in the stabilization process of mRNAs as Cyt b, ND5, ND6, and Cyt b mRNAs [15].

The predominant subcellular location of SncmtRNA and ASncmtRNAs shifts from mitochondria towards the nucleus. ASncmtRNAs knockdown (ASK) by antisense oligonucleotide (ASO-1537S) targets the single-stranded loop region, causing a dicer-mediated release of hsa-miR-4485. The level of nuclear miRNAs and the mediated release of hsa-miR-4485 shows a preference for hsa-miR-5096 and hsa-miR-3609, which are induced by ASK. Next, an inhibited translation of five important cell cycle proteins (Cyclin B1, Cyclin D1, CDK1, CDK4, and also Survivin) results in an induced growth inhibition of cells during breast cancer [16]. This is also shown in ASK-induced bladder cancer cell death and results in an inhibition of tumor growth [17]. In summary, this indicates that ASncmtRNAs could be potential therapeutic targets for breast cancer and bladder cancer in future. Unforeseen, this missing link as the regulatory property of mt-lncRNAs in human diseases and its clinical application is of great value, as reported [18]. Recently, it was reported that mt-lncRNAs contribute to more than 70% of the total amount of lncRNA pool in the human left ventricle area. The high proportion of mt-lncRNAs seems to have originated from the heart. Briefly, it is reported that mitochondrial lncRNA, uc022bqs.1, or long intergenic non-coding RNA predicting CARdiac remodeling (LIPCAR), was downregulated early after myocardial infarction but upregulated during later stages. The levels of LIPCAR identified in patients developing cardiac remodeling and leading independently to other risk markers associated with cardiovascular deaths in future [19].

In extensive studies, the exportation and circulation mechanisms of lncRNAs need to be explored to clarify the complex function of the mitochondrial role of LIPCAR in regulating biological pathways in processes such as oxidative phosphorylation (OXPHOS). Thus, the potential functional role of lncRNA in mitochondrial biology, as shown in Figure 1 and Table 1, needs to be highlighted based on the last two decades of improved understanding of high-throughput deep sequencing technologies.

MD forms an inherited heterogeneous group of genetic disorders in high energy-demanding tissues and organs. However, the use of targeted mitochondrial DNA sequencing (mtDNA-Seq), whole exome sequencing (WES), and whole genome sequencing (WGS) improved the diagnostic yield in the search for disease-causing variants in MD. It has recently become clear that large-scale genetic research on the pathophysiology of tissues and cell lines does not sufficiently map the underlying complexity and heterogeneity in relation to these disease-causing variants or mutation levels in the case of homoplasmic or heteroplasmic mtDNA mutations. Tissues are made up of large numbers of cells that can differ greatly from one another, and especially the differences in mtDNA copies and mutation percentages per cell make this level of understanding a little bit foggy or misunderstood in the transmission mechanism during MD. The same applies to other diseases, such as tumors, which consist of cells with different genetic abnormalities, a different transcriptome and epigenome, and, therefore, a different malignancy.

Mitochondria are a dual inherited genetic organelle, with genetic components that contribute to mitochondrial function and structure originating from two different sources. One is mtDNA (maternal), and the other is nuclear DNA or nDNA (maternal and paternal). More than 1500 nDNA-inherited genes are involved in the processing of the structural proteins and constructive protein complexes regulating the homeostatic balance of mitochondria in specific biological processes such as DNA replication, or transcriptional and translational maintenance [20]. A few studies have reported RNA-seq as a screening tool to identify splice variants in inherited diseases. The success of deep transcriptome sequencing to increase the mutation detection rate in undiagnosed mitochondrial diseases resulted in the identification of causative, non-exonic regulatory variants in an additional 10% of cases [21].

Standard bulk analyses on DNA genomics or RNA whole transcriptomics technologies have limited sensitivity when it comes to cell type and cell subtype-specific mutations and expression profiles. For complex, cell-level molecular analyses, new technologies are developed that offer the possibility to determine the transcriptome (Single-Cell-RNA-seq), epigenome, as open DNA chromatin sequencing (Single-Cell-ATAC-seq), or immune cell profile (Single-Cell-Immune-Profiling-seq), in individual cells or cell nuclei (single-cell and single-nuclei levels). Furthermore, spatial transcriptomics offers not only cellular and regional information on RNA expression in fresh frozen (FF) biopsies or even Formalin-Fixed Paraffin-Embedded (FFPE) tissue. Single-cell omics technologies are set to increase in sensitivity and resolution, which could detect rare mtDNA mutations as well as identify novel lncRNAs that regulate mitochondrial function. These novel technologies will help accelerate understanding of the presence, features, and roles of ncRNAs in the pathophysiology of MDs and reveal important new drug targets for presently incurable primary MDs as well as secondary mitochondrial dysfunction.

LncRNAs have been reported to directly and indirectly affect mitochondrial biology through signaling across complex ncRNA networks (Table 1). The lack of understanding of lncRNA trafficking remains a fundamental knowledge gap [22]. This commentary reports on the fundamental status and the role of lncRNAs leading to mitochondrial dysfunction relating to several mitochondria-related human diseases, such as aging, neurodegeneration, rare cancers, and cardiac and cardiovascular diseases. More interest in lncRNAs needs to be stimulated by the development of high-throughput multiomics approaches and state-of-the-art technologies (e.g., whole genome sequencing (WGS), RNA-sequencing (RNA-seq), and active ribosomal sequencing (Ribo-seq) as LR sequencing methods. LR sequencing will contribute to detecting deep intronic, single nucleotide variants (SNVs), insertions and deletions (Indels), tandem repeats (TRs), and structural variants (SVs) [23]. These new advanced-sensitive and cost-effective deep sequencing technologies can be used to unravel the interaction between specific cell tissue expression of the general mitochondrial long ncRNAs (mt-lncRNAs), which suggests that they may take part in mitochondria-related inherited rare diseases.

LR-PacBio (HiFi long-read sequencing), shown as full-length transcriptomic data of two extra novel lncRNAs, is described in relation to human mitochondrial DNA sequencing, listed as MDL1 and MDL1AS [24]. The MDL1 is the counterpart or listed as the antisense sequence of the D-loop region and tRNAPro gene counterpart. This is also part of its antisense transcript sequence named MDL1AS. However, neither of these two novel lncRNAs has been studied for mitochondrial and cellular localization nor for expression levels in mitochondria in humans. In general, the advantage of using HiFi-resolution LR sequencing is to discover the complete spectrum of genetics as structural variation in most of the important listed deep sequencing applications elucidating isoform diversity in progressive human disease development.

Newly available tools for mitochondrial quantification are reported to identify differences in mitochondrial morphology, resulting in a deeper understanding of these complex disease-causing networks [25]. Recently, new guidelines for mitochondrial RNA analysis have been reported to study the role of anterograde- and retrograde-ncRNAs in the regulation of specific mitochondrial genes and their functional biological pathways [26]. These guidelines show that using standardized protocols for the isolation, purification, and quantification of mitochondria is required to reduce the level of variability and discrepancy in the previously listed mitochondrial results in the area of genetics and human diseases.

Lately, two independent studies reported a unique regulation of the two lncRNAs, Kcnq1ot1 and Cerox1, as novel mRNA targets [27,28]. The lncRNA Kcnq1ot1 plays a role in the regulation of ATP synthase in a diabetic heart via the miR-387a/mt-ATP6 upregulated pathway and increased activity of complex V. Additionally, it plays a role in Cerox1 catalysis, complex I activity in the oxidative phosphorylation (OXPHOS), and miR-488-3p as a novel post-transcriptional regulator. The crosstalk of the lncRNAs classified as mt-lncRNAs or the mitochondria-associated lncRNAs are localized in human cells. The mitochondria-associated lncRNAs are present in the nuclear genome and not located in the mitochondria. These kinds of associated RNAs regulate mitochondrial functions via the nucleus. However, the mt-lncRNAs, metastasis-associated lung adenocarcinoma transcript 1, and MALAT1 are involved in the epigenetic and metabolic reprogramming in HepG2 HCC cells [29]. Survival Associated Mitochondrial Melanoma-Specific Oncogenic Non-Coding RNA (SAMMSON) modulates the levels of expression of mt-DNA-encoded OXPHOS complex subunits via p32 in skin cancer cells [30,31]. Both mt-lncRNAs are shuttling between the mitochondria and the nucleus or in other compartments of the human cell [32,33]. H1RNA, RMRP, and hTERC are mostly reported in an unknown role as lncRNA originated in the nucleus and the association with mitochondria [22,34,35].

**Table 1 ncrna-10-00057-t001:** The location and classification of lncRNAs implicated in mitochondrial function and disease. The reference content was compiled by searching PubMed and manually filtering on main keywords such as long non-coding RNA or mitochondria-related human diseases and based on the last two decades of improved understanding in high-throughput deep sequencing technologies.

lncRNA	Origin	Location	Function	Disease	nDNA	mtDNA	Strand	Ref
AsncmtRNAs	Human	Mitochondria	Unknown	Unknown		x	L-strand	[22,24]
Cerox1	Mouse	Nuclear	Knockdown decrease in Complex I and IV	Cancer	x	Associated		[22,28]
GAS5	Human	Nuclear	TCA disorganization due to retardation process of the TCA	Cancer	x	Associated		[35]
lncCytb	Human	Mitochondria	Stabilize and regulate gene expression of mt-Cytb	Unknown		x	L-strand	[11,22]
lncND5	Human	Mitochondria	Stabilize and regulate gene expression of mt-ND5	Unknown		x	L-strand	[11,22]
lncND6	Human	Mitochondria	Stabilize and regulate gene expression of mt-ND6	Unknown		x	H-strand	[11,22]
H1 RNA	Human	Nuclear	Unknown role in RNA metabolism	Unknown	x	Associated?		[22,34]
hTERC	Human	Nuclear	Telomerase and RNA component, processed and transported towards cytosol as unknown role?	Unknown	x	Associated?		[22,34]
LIPCAR	Human	Mitochondria, Cytoplasm	Chimeric mtDNA-encoded transcript	Coronary heart failure		x	L-strand	[11,22]
lncFAO	Mouse	Nuclear	Binding to HADHB to promote fatty acid oxidation	Unknown	x	Associated		[35]
Kcnq1ot1	Mouse	Nuclear	Regulation ATP synthase, Complex V	Cancer and Cardiomyopathy	x	Associated		[22,24]
MALAT1	Human	Nuclear	CpG methylation and mtDNA-encoded ETC complex	Cancer and metastasis	x	Associated		[22,33]
MDL1AS	Human	Mitochondria	Unknown role	Unknown		x	L-strand	[22,24]
MDL1	Human	Mitochondria	Unknown role	Unknown		x	H-strand	[22,24]
RMRP	Human, Mouse	Nuclear	Pre-rRNA processing of 5.8S RNA	Unknown	x	Associated?		[22,34]
SAMMSON	Human	Nuclear, Cytoplasm	P32 target and mtDNA-encoded ETC complex	Severe melanomas	x	Associated?		[22,33]
SncmtRNA	Human	Mitochondria	Unknown	Unknown		x	H-strand	[22,33]

## 4. Future Perspectives

A number of characterized lncRNAs, including LIPCAR and Kcnq1ot1, are established as playing a role in cardiovascular disease biology. Others, like MALAT1, SAMMSON, and Cerox1, are implicated in the development of cancers. However, many identified lncRNAs, including mt-lncRNAs, remain poorly characterized and could represent modifiers of heart function and cardiovascular disease. Recent advances in multiomics approaches, such as single-cell RNA-seq, are now being used to investigate the regulatory network of mitochondrial and nuclear-derived lncRNAs. Highly sensitive omics technologies will pave the way to improve understanding of the dynamics and role of lncRNA in disease and reveal new RNA drug targets [18,36].

In summary, emphasizing the scientific impact of lncRNAs in regulating the structure and function of mitochondria as the energy powerhouse of the cell could be a game-changer in disease genetics and reveal novel targeted therapies for untreatable MD cases and for chronic diseases associated with secondary mitochondrial dysfunction.

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
