# Peer review of "LNC-ing Genetics in Mitochondrial Disease"

_ncrna, 2024, doi:10.3390/ncrna10060057_

Round 1
Reviewer 1 Report
Comments and Suggestions for Authors
This interesting commentary manuscript by Dr. Kamps and Robinson explores the emerging role of long non-coding RNAs (lncRNAs) in mitochondrial diseases (MD). It provides an overview of how these diseases are caused by dysfunctions in mitochondrial DNA (mtDNA) and nuclear DNA (nDNA) and investigates the potential of lncRNAs to contribute to the diagnosis, prognosis, and treatment of MD.
1. The authors categorize lncRNAs into mitochondrial and nuclear classes based on gene location. However, Table 1 presents some inconsistencies. For instance, the lncRNA "LIPCAR" is listed under "plasma" in the location category, while "SAMMSON" is listed under "cytoplasm". This classification needs clarification to avoid confusion.
2. While the review focuses on long non-coding RNAs, it would be beneficial to briefly discuss whether short non-coding RNAs (<200 nt) play a role in mitochondrial diseases. Additionally, the authors should explain their rationale for focusing on lncRNAs specifically, which would enhance the context and scope of the review.
3. The "Future Perspectives" section is concise but could be expanded. Elaborating on specific research priorities and addressing unanswered questions would strengthen the conclusion and provide readers with a clearer roadmap for future investigations in this field.
4. On line 150, the term "mt-lncRNA" should be pluralized to "mt-lncRNAs" for consistency with its usage elsewhere in the manuscript.
Author Response
Response to Reviewers R1 ncrna-3108022
We thank the editor and peer reviewers for the opportunity to have our work considered for publication in NcRNA and for your time spent reviewing our article. We have addressed the peer reviewer comments and provided a point-by-point response below. Thank you.
Our responses are in purple.
Reviewer 1:
This interesting commentary manuscript by Dr. Kamps and Robinson explores the emerging role of long non-coding RNAs (lncRNAs) in mitochondrial diseases (MD). It provides an overview of how these diseases are caused by dysfunctions in mitochondrial DNA (mtDNA) and nuclear DNA (nDNA) and investigates the potential of lncRNAs to contribute to the diagnosis, prognosis, and treatment of MD.
- The authors categorize lncRNAs into mitochondrial and nuclear classes based on gene location. However, Table 1 presents some inconsistencies. For instance, the lncRNA "LIPCAR" is listed under "plasma" in the location category, while "SAMMSON" is listed under "cytoplasm". This classification needs clarification to avoid confusion. Thank you for pointing this out. We have amended the categorization of LIPCAR to cytoplasm in Table 1.
- While the review focuses on long non-coding RNAs, it would be beneficial to briefly discuss whether short non-coding RNAs (<200 nt) play a role in mitochondrial diseases. Additionally, the authors should explain their rationale for focusing on lncRNAs specifically, which would enhance the context and scope of the review. This is a great point, thank you. We have now acknowledged the presence of mtDNA-encoded microRNAs and explained our focus on lncRNAs in lines 80-85.
- The "Future Perspectives" section is concise but could be expanded. Elaborating on specific research priorities and addressing unanswered questions would strengthen the conclusion and provide readers with a clearer roadmap for future investigations in this field. Thank you. The Future Perspectives as well as Summary statement have been heavily revised.
4. On line 150, the term "mt-lncRNA" should be pluralized to "mt-lncRNAs" for consistency with its usage elsewhere in the manuscript. Thank you. We have made this amendment.

Reviewer 2 Report
Comments and Suggestions for Authors
Thanks to the authors for this attempt at a review article on one of the most difficult and challenging conundrum in medical genetics: the high variability in mitochondrial disease expression for the same nuclear or mitochondrial pathogenic variant. In the present version, this remains an attempt and much work remains to be done before it can be considered a valuable addition to pre-existing articles on the same topic.
But I encourage you to do it, you already have a good part of the material to do it, only the architecture of the article needs to be improved.
So please help readers navigate this maze while being more methodical and didactic from the introduction( not solely in the abstract). Mitochondrial diseases can be caused by pathogenic variants encoded in the nuclear genome as well as the mitochondrial genome. For mitochondrial genome variants, they can be found in either homoplasmic or heteroplasmic state. But this distinction is not enough to explain the great variability of MD expression. For example, Leber's hereditary optic neuropathy is caused by a homoplasmic pathogenic variant of the mitochondrial genome and there remains extreme symptomatic variability among carriers of this variant.
There is therefore much to discover to explain this variability, and it is here that we must introduce the saga of non-coding variants with all their diversity linked to their size. You will be on LNC. They can come from the mitochondrial genome or the nuclear genome and LNCs from these two origins can interact with each other as well as with other ncvs, notably miRNAs.
We must now consider some fundamental points.
I find your genomic approach to identifying disease mediators to be a bit outdated. The current modern approach is WGS in trio first and makes most sense for searching for lnc variants of significant pathological interest. All the other techniques you describe can sometimes be used to decipher certain difficulties in interpreting WGS and they help advance bioinformatics to alleviate these interpretation ambiguities. This approach also has the advantage of simplifying your description.
If you want you can always cite WES as a trio but as an almost historical method in this hunt for lncs
Please to rewrite the 3rd chapter secondary mitochondrial dysfunction and LncRNAs, it can be clearer. I read some papers you give in reference and I found their abstract much clearer than the manner you report them.
Minor criticisms: line 54 "nearly 100,000 disease-associated lncRNAs" is a strange approximation, not very serious and a vague extrapolation of the contents of reference 4
Author Response
Response to Reviewers R1 ncrna-3108022
Our responses are in purple.
We thank the editor and peer reviewers for the opportunity to have our work considered for publication in NcRNA and for your time spent reviewing our article. We have addressed the peer reviewer comments and provided a point-by-point response below. Thank you.
Reviewer 2:
Thanks to the authors for this attempt at a review article on one of the most difficult and challenging conundrum in medical genetics: the high variability in mitochondrial disease expression for the same nuclear or mitochondrial pathogenic variant. In the present version, this remains an attempt, and much work remains to be done before it can be considered a valuable addition to pre-existing articles on the same topic. But I encourage you to do it, you already have a good part of the material to do it, only the architecture of the article needs to be improved. So please help readers navigate this maze while being more methodical and didactic from the introduction (not solely in the abstract). Mitochondrial diseases can be caused by pathogenic variants encoded in the nuclear genome as well as the mitochondrial genome. For mitochondrial genome variants, they can be found in either homoplasmic or heteroplasmic state. But this distinction is not enough to explain the great variability of MD expression. For example, Leber's hereditary optic neuropathy is caused by a homoplasmic pathogenic variant of the mitochondrial genome and there remains extreme symptomatic variability among carriers of this variant.
There is therefore much to discover to explain this variability, and it is here that we must introduce the saga of non-coding variants with all their diversity linked to their size. You will be on LNC. They can come from the mitochondrial genome or the nuclear genome and LNCs from these two origins can interact with each other as well as with other ncvs, notably miRNAs. We must now consider some fundamental points. I find your genomic approach to identifying disease mediators to be a bit outdated. The current modern approach is WGS in trio first and makes most sense for searching for lnc variants of significant pathological interest. All the other techniques you describe can sometimes be used to decipher certain difficulties in interpreting WGS and they help advance bioinformatics to alleviate these interpretation ambiguities. This approach also has the advantage of simplifying your description. If you want you can always cite WES as a trio but as an almost historical method in this hunt for lncs. Thank you. We have heavily revised this introduction and the genomic approach paragraph.
Please to rewrite the 3rd chapter secondary mitochondrial dysfunction and LncRNAs, it can be clearer. I read some papers you give in reference and I found their abstract much clearer than the manner you report them. Thank you. We have heavily revised this paragraph.
Minor criticisms: line 54 "nearly 100,000 disease-associated lncRNAs" is a strange approximation, not very serious and a vague extrapolation of the contents of reference 4. Thank you. We have now amended this sentence.

Round 2
Reviewer 2 Report
Comments and Suggestions for Authors
I was waiting with interest for your answers to my comments, they have improved a few points but overall the text does not reach the level required for a scientific publication. The table is the best thing in the article, but even if you do not have made métalanalysis, you should have given a few indications on the methodology for constructing it (keywords, data base consulted).
Te introduction remains unacceptable with many vague or false définitions (3 between lines 61 and 65).
Finally, you did not explain clearly how the single cell omics technologies will improve our understanding.
Author Response
Response to Reviewer 2 ncrna-3108022
We appreciate the additional time spent reviewing our manuscript, and feedback from the reviewer.
I was waiting with interest for your answers to my comments, they have improved a few points but overall the text does not reach the level required for a scientific publication. The table is the best thing in the article, but even if you do not have made métalanalysis, you should have given a few indications on the methodology for constructing it (keywords, data base consulted).
This article was not intended to be a metanalysis, as the reviewer indicates. We have now included a line in the legend for Table 1 as to how the manuscripts in Table 1 were identified.
Te introduction remains unacceptable with many vague or false définitions (3 between lines 61 and 65).
The introduction has been heavily revised, as indicated in the highlighted sections.
Finally, you did not explain clearly how the single cell omics technologies will improve our understanding.
Again, the single cell omics section has been revised to include how it may impact our knowledge of mtlncRNA as genetic mediators of disease.
Round 3
Reviewer 2 Report
Comments and Suggestions for Authors
Thank you, the revised manuscript is now much clearer and will be useful to readers.